

# Influencing factors of cardiac valve calcification (CVC) in patients with chronic kidney disease and the impact of CVC on long-term prognosis: a single-center retrospective study

Ju Wang, Jianping Xiao, Ruifeng Wang and Deguang Wang

Department of Nephrology, The Second Affiliated Hospital of Anhui Medical University, Hefei, Anhui Province, China

## ABSTRACT

**Objective**. To investigate the effect of cardiac valve calcification (CVC) on the prognosis of patients with chronic kidney disease (CKD).

**Methods**. A total of 343 CKD patients were retrospectively analyzed, and divided into two groups according to the presence or absence of cardiac valve calcification. All patients were followed until death, loss to follow-up, or the end point of the study (December 2021).

**Results**. The incidence of CVC among the 343 CKD patients was 29.7%, including 21 cases of mitral valve calcification, 63 cases of aortic valve calcification, and 18 cases of mitral valve combined with aortic valve calcification. The incidence of CVC in CKD stages 1–2 was 0.3%, 5.2% in CKD stages 3–4, and 24.2% in CKD stage 5 ($P < 0.05$). Advanced age, higher serum albumin, higher cystatin C and lower uric acid levels were all associated with a higher risk of CVC. After six years of follow-up, 77 patients (22.4%) died. The causes of death were cardiovascular and cerebrovascular diseases in 36 cases (46.7%), infection in 29 cases (37.7%), gastrointestinal bleeding in nine cases (11.7%), and "other" in the remaining three cases (3.9%). A Kaplan Meier survival analysis showed that the overall survival rate of patients with CVC was lower than that of patients without CVC.

**Conclusion**. The incidence of CVC, mainly aortic calcification, is high in patients with CKD. Advanced age, higher serum albumin and higher cystatin C levels were associated with a higher risk of CVC. Hyperuricemia was associated with a lower risk of CVC. The overall survival rate of patients with CVC was lower than that of patients without CVC.

Corresponding author
Deguang Wang,
wangdeguang@ahmu.edu.cn

## INTRODUCTION

Chronic kidney disease (CKD) patients often have comorbid complications such as anemia, malnutrition, mineral bone metabolism disorders, cardiovascular and cerebrovascular diseases and other chronic complications (*Kidney Disease: Improving Global Outcomes (KDIGO) Anemia Work Group, 2012*; *Carrero et al., 2018*; *Hou, Lu & Lu, 2018*; *Go et al.,*

*2004*; *Lee et al., 2010*). Abnormal mineral bone disorder (MBD) is one of the most common complications in maintenance hemodialysis (MHD) patients, which manifests as calcium and phosphorus metabolism disorder or secondary hyperparathyroidism (*Gimba et al., 2018*). The Global Organization for Improving Outcomes in Kidney Disease (KDIGO) guidelines recommend that CKD patients in stages 3–5 (CKD3-5) with vascular or valvular calcification be considered the highest risk group for cardiovascular disease (*Kidney Disease: Improving Global Outcomes (KDIGO) CKD-MBD Work Group, 2009*). Cardiac valve calcification (CVC) can lead to cardiac conduction dysfunction, myocardial ischemia or infarction, valve insufficiency, congestive heart failure, and other complications, increasing the risk of cardiovascular death (*Demer & Tintut, 2008*; *Bai et al., 2022*).

Hyperphosphatemia is an important cause of increased vascular calcification in patients with CKD, which also leads to increased mortality (*Wang et al., 2001*). The exact mechanism of calcification caused by CKD, however, has not yet been identified (*Cozzolino et al., 2005*). Small sample studies have shown that renal function loss is faster and the incidence of valve calcification is higher in patients with CKD5 hyperphosphatemia without dialysis, and serum phosphorus level can be used as an independent predictor of total calcification score (*Lezaic et al., 2009*). In patients with continuous ambulatory peritoneal dialysis, hyperphosphatemia accelerated the rate of calcification, and patients with inflammation and malnutrition had an increased prevalence of CVC (*Wang et al., 2001*).

There are few studies on the effect of cardiac valve calcification on the long-term prognosis of patients with CKD. This study explored the incidence of cardiac valve calcification in patients with CKD and the risk factors for CVC, seeking to understand both the correlation and impact of CVC on all-cause mortality in patients with CKD. The results of this study provide a theoretical basis for improving the prognosis of patients with CKD.

## METHODS

### Participants

A total of 343 CKD patients hospitalized in the Department of Nephrology at the Second Affiliated Hospital of Anhui Medical University between January 2015 and March 2015 were retrospectively analyzed. The diagnosis and staging criteria of CKD used in this study were based on those outlined in the K/DOQI guidelines (*National Kidney Foundation, 2003*). Patients were excluded from the study if they were: <18 years of age; had an acute infection within the last month; had CKD combined with a malignant tumor, acute renal failure (including acute exacerbation based on chronic kidney disease), or congenital heart disease; or had ever had surgery for heart valve disease. The flow chart of patient enrollment is shown in Fig. 1. This study was approved by the Ethics Committee of the Second Affiliated Hospital of Anhui Medical University (project number: YJ-YX2017-004) and written, informed consent was obtained from all study participants.

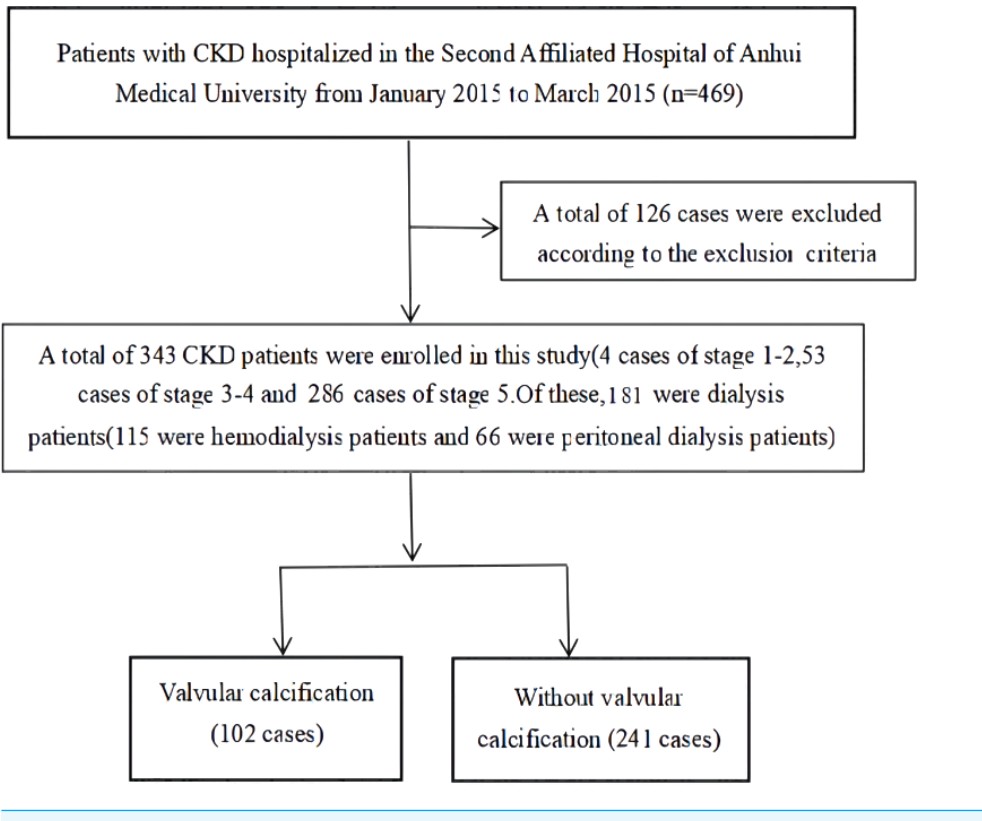

**Figure 1** Filtering flow chart.

## Clinical data and examination indicators

General demographic characteristics, sleep duration, primary diseases, complications, dialysis-related conditions and medication use were collected for each patient.

Hemoglobin (Hb), fasting plasm glucose (FPG), serum albumin (Alb), total cholesterol (TC), and triglyceride (TG) were also recorded for each patient. Laboratory tests, such as serum creatinine (Scr), blood urea nitrogen (BUN), calcium (Ca), phosphorus (P), intact parathyroid hormone (iPTH) and 25-hydroxyvitamin D3[25(OH)D3] were collected. Lateral abdominal radiography, cervical vascular color Doppler ultrasound and cardiac color Doppler ultrasound were analyzed for each study participant to determine the presence of CVC.

Serum triglyceride glucose product index (TyG) was calculated using the following formula: TyG = Ln[fasting triglyceride (mg/dl) ∗ fasting glucose (mg/dl)/2] (*Simental-Mendia, Rodriguez-Moran & Guerrero-Romero, 2008*).

When serum Alb < 40 g/L, the following formula was used for correction: corrected calcium (mmol/L) = total serum calcium (mmol/L)−0.02 ∗ [Alb (g/L)-40 g/L] (*Phillips & Pain, 1977*).

## Diagnostic criteria

Patients were considered to have CVC if a strong echo >1 mm was found on one or more heart valves or annuli by color Doppler echocardiography (*Wong, Tei & Shah, 1983*).

Patients were then divided into a CVC group and a non-CVC group according to the presence or absence of CVC.

## Follow-up and study end points

All patients were followed until death, loss to follow-up, or the end point of the study (December 2021). The primary endpoint was all-cause mortality.

## Statistical analysis

SPSS 26.0 software was used for the statistical processing of data. Quantitative data with normal distribution were represented as (mean ± standard deviation), and quantitative data with skewed distribution were represented as *M(1/4, 3/4)*. The independent sample *t*-test or rank sum test was used to compare the fixed volume data between the two groups, and the $\chi 2$ comparison was used to compare the count data. The binary logistic regression method was used to analyze the risk factors related to CVC. A Kaplan–Meier survival curve (Log-rank test) was used to compare the differences in survival rates between groups. $P < 0.05$ was considered statistically significant.

## RESULTS

A total of 343 CKD patients were included in this study, including 199 males and 144 females, with an average age of 54.3 ± 15.9 years. There were 162 non-dialysis patients and 181 dialysis patients (115 hemodialysis patients and 66 peritoneal dialysis patients). The cause, or primary disease leading to CKD among the study participants were as follows: chronic glomerulonephritis (47.9%), hypertensive nephropathy (20.4%) and diabetic nephropathy (17.5%), polycystic kidney (5.0%), gouty nephropathy (5.0%), obstructive nephropathy (1.8%), lupus nephritis (1.2%), purpura nephritis (0.6%), and other (0.6%).

Among the 343 included CKD patients, the incidence of CVC was 29.7% (102 cases), including 21 cases of mitral valve calcification (20.6%), 63 cases of aortic valve calcification (61.8%), and 18 cases of mitral valve combined with aortic valve calcification (17.6%).

Patient age, serum Alb, cystatin C, TG, TyG index, CRP, and serum phosphorus levels, and the incidence of abdominal aortic calcification were all significantly higher in the CVC group than in the non-CVC group ($P < 0.05$). and Ddiastolic pressure, Hb, serum uric acid and left ventricular ejection fraction (LVEF) levels were lower in the CVC group than in the non-CVC group. The differences between the groups were statistically significant ($P < 0.05$). The proportion of patients with chronic glomerulonephritis in the CVC group was lower than in the non-CVC group ($P < 0.05$) and the mortality rate in the CVC group was higher than that in the non-CVC group (29.4% *vs.* 19.5%, $P < 0.05$). There were no significant differences in corrected blood calcium, iPTH and 25(OH)D3 levels between the two groups (Table 1).

The comparison of the incidence of valve calcification in different stages of CKD (CKD1-2, CKD3-4, CKD5) is shown in Fig. 2. The incidence of CVC was 0.3% in CKD1-2,

**Table 1 Comparison of clinical data between cardiac valve calcification group and non calcification group.**

| Characteristics | VC group (n = 102) | Non-CVC group (n = 241) | t/χ² | P-value |
|---|---|---|---|---|
| Male (n (%)) | 60 (58.8) | 139 (57.7) | 0.039 | 0.844 |
| Age (years, $\bar{x} \pm s$) | 60.1 ± 15.1 | 52.5 ± 15.5 | 4.130 | <0.001 |
| BMI (kg/m2, $\bar{x} \pm s$) | 23.1 ± 4.9 | 22.3 ± 4.8 | 1.385 | 0.167 |
| Sleep time (hour/day, $\bar{x} \pm s$) | 6.2 ± 2.5 | 6.6 ± 1.7 | −1.786 | 0.075 |
| Smoking (n (%)) | 12 (11.7) | 33 (13.7) | 0.234 | 0.629 |
| Alcohol consumption (n (%)) | 11 (10.7) | 37 (15.3) | 1.243 | 0.265 |
| Diabetes mellitus (n (%)) | 24 (23.5) | 42 (17.4) | 1.717 | 0.190 |
| Hypertension (n (%)) | 75 (73.5) | 176 (73.0) | 0.009 | 0.924 |
| Dialysis patients (n (%)) | 53 (51.9) | 128 (53.1) | 0.038 | 0.845 |
| CKD stage (n (%)) | | | 0.566 | 0.798 |
| CKD1-2 | 1 (1) | 3 (1.2) | | |
| CKD3-4 | 18 (17.6) | 35 (14.6) | | |
| CKD5 | 83 (81.3) | 203 (84.2) | | |
| Systolic blood pressure (mmHg, $\bar{x} \pm s$) | 143.3 ± 22.5 | 144.6 ± 21.8 | −0.507 | 0.613 |
| Diastolic pressure (mmHg, $\bar{x} \pm s$) | 84.5 ± 12.6 | 88.6 ± 14.8 | −2.382 | 0.018 |
| Hemoglobin (g/L, $\bar{x} \pm s$) | 77.0 ± 37.6 | 90.3 ± 27.1 | −3.666 | <0.001 |
| Alb (g/L, $\bar{x} \pm s$) | 38.0 ± 12.6 | 33.7 ± 7.4 | 3.773 | <0.001 |
| Fasting plasma glucose (mmol/L, M (1/4, 3/4)) | 4.7 (4.0, 6.0) | 5.0 (4.0, 6.0) | −1.638 | 0.101 |
| Cystatin C (mg/L,M (1/4, 3/4)) | 6.0 (4.0, 9.1) | 4.3 (3.0, 7.1) | −2.785 | 0.005 |
| Uric acid (μmol/L, $\bar{x} \pm s$) | 368.9 ± 156.9 | 444.6 ± 158.5 | −3.913 | <0.001 |
| Total cholesterol (mmol/L,M (1/4,3/4)) | 3.8 ± 1.5 | 4.2 ± 2.8 | −1.556 | 0.121 |
| TG (mmol/L,M (1/4, 3/4)) | 7.4 ± 13.1 | 2.0 ± 3.6 | 5.739 | <0.001 |
| TyG index | 8.7 (8.2, 9.7) | 8.4 (8.0, 9.1) | −2.670 | 0.008 |
| CRP (mg/L, M (1/4, 3/4)) | 12.0 (3.1, 29.5) | 5.0 (2.0, 11.0) | −3.518 | <0.001 |
| Corrected serum calcium (mmol/L,M (1/4, 3/4)) | 2.1 (1.8, 2.2) | 2.1 (2.0, 2.2) | −1.951 | 0.051 |
| Phosphorus (mmol/L, $\bar{x} \pm s$) | 1.9 ± 0.6 | 1.9 ± 0.7 | −0.655 | 0.513 |
| ALP (U//L,M (1/4, 3/4)) | 95.0 (59.5, 196.5) | 81.0 (60.0, 121.0) | −0.888 | 0.374 |
| iPTH (pg/ml,M (1/4, 3/4)) | 167.0 (52.0, 615.0) | 214.0 (118.0, 453.0) | −1.060 | 0.289 |
| 25 (OH)D3 (μg/L,M (1/4, 3/4)) | 7.0 (3.3, 12.9) | 6.0 (4.0, 14.9) | −1.013 | 0.311 |
| PAP (mmHg, M (1/4,3/4)) | 27.5 (21.0, 38.0) | 28.0 (21.0, 33.0) | −0.369 | 0.712 |
| LVEF (%, M (1/4, 3/4)) | 60.0 (57.2, 64.0) | 62.0 (58.7, 65.0) | −1.973 | 0.049 |
| Etiology of CKD | | | | |
| Chronic glomerulonephritis (n (%)) | 42 (43.8) | 120 (61.9) | 8.538 | 0.003 |
| Hypertensive nephropathy (n (%)) | 29 (30.2) | 40 (20.6) | 3.257 | 0.07 |
| Diabetic nephropathy (n (%)) | 25 (26.0) | 34 (17.5) | 2.874 | 0.09 |
| Calcification of abdominal aorta (n (%)) | 29 (28.4) | 40 (16.6) | 6.245 | 0.012 |
| Use of phosphorus binding agent (n (%)) | 61 (59.8) | 91 (37.7) | 3.744 | 0.053 |
| Calcium phosphate binder (n (%)) | 47 (46.1) | 86 (35.7) | | |
| Aluminum-phosphorus binder (n (%)) | 5 (4.9) | 0 (0) | | |
| Non-calcium aluminum and non-phosphorus binder (n (%)) | 9 (8.8) | 5 (2.1) | | |

**Table 1** (*continued*)

| Characteristics | VC group (n = 102) | Non-CVC group (n = 241) | t/χ² | P-value |
|---|---|---|---|---|
| Take active vitamin D or calcium simulants (*n* (%)) | 25 (24.5) | 48 (19.9) | 0.902 | 0.342 |
| Mortality rate (*n* (%)) | 30 (29.4) | 47 (19.5) | 4.043 | 0.044 |
| Cause of death | | | | |
| Cardiovascular and cerebrovascular diseases (*n* (%)) | 12 (40.0) | 24 (51.0) | 0.900 | 0.343 |
| Infection (*n* (%)) | 14 (46.7) | 15 (31.9) | 0.253 | 0.615 |
| Gastrointestinal bleeding (*n* (%)) | 3 (10.0) | 6 (12.7) | 0.136 | 0.713 |
| Other (*n* (%)) | 1 (3.3) | 2 (4.2) | 0.042 | 0.838 |

**Notes.**
CVC, cardiac valve calcification; BMI, body mass index; Alb, albumin; TG, triglyceride; CRP, c-reactive protein; ALP, alkaline phosphatase; iPTH, intact parathyroid hormone; 25 (OH)D3, 25-hydroxyvitamin D3; PAP, pulmonary artery pressure; LVEF, left ventricular ejection fraction.

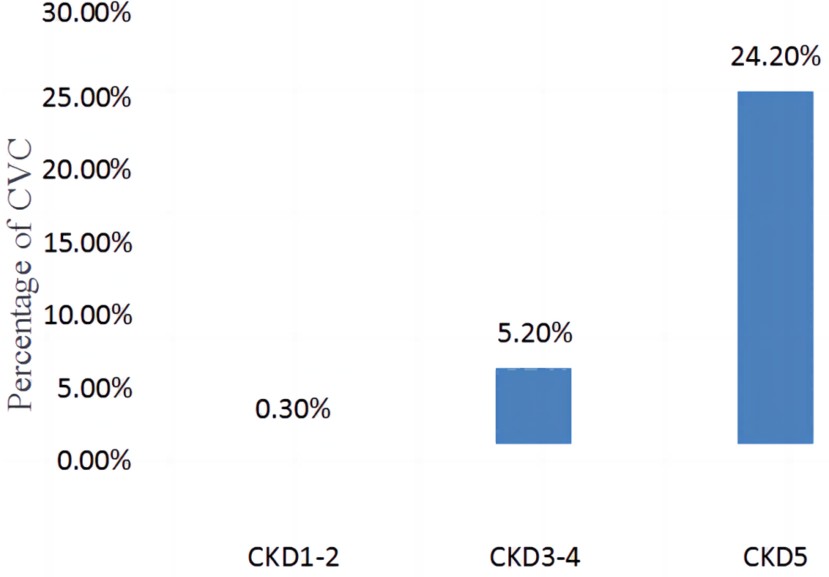

**Figure 2** Comparison of the incidence of CVC in different CKD stages (CKD1-2, CKD3-4, CKD5).

5.2% in CKD3-4, and 24.2% in CKD5, with statistically significant differences between the groups ($\chi2 = 525.636$, $P < 0.001$).

A binary logistic regression analysis was performed with the presence of CVC as the dependent variable. Patient age, diastolic blood pressure, Hb, Alb, cystatin C, serum uric acid, TG, TyG index, CRP, corrected serum calcium, LVEF, abdominal aortic calcification and the use of phosphorus binder were input as the independent variables. Entry method was selected. The results showed that advanced patient age, higher serum albumin levels and higher cystatin C levels were all associated with a higher risk of CVC. Hyperuricemia was associated with a lower risk of CVC (Table 2).

After six years of follow-up, 77 patients (22.4%) died, 12 patients(3.5%) received renal transplantation, and 25 patients (7.2%) were lost to follow-up. The causes of death were

**Table 2  Multiple linear regression analyses affecting the occurrence of cardiac valve calcification in patients with CKD.**

| Characteristics | Wald | OR | 95% CI | P-value |
|---|---|---|---|---|
| Age (years) | 4.687 | 1.029 | 1.003–1.057 | 0.030 |
| Diastolic pressure (mmHg) | 1.731 | 0.982 | 0.955–1.009 | 0.188 |
| Hb (g/L) | 0.055 | 0.997 | 0.976–1.020 | 0.815 |
| Alb (g/L) | 6.621 | 1.117 | 1.027–1.215 | 0.010 |
| Cystatin C (mg/L) | 12.211 | 1.275 | 113–1.462 | <0.001 |
| Uric acid (μmol/L) | 5.212 | 0.996 | 0.992–0.999 | 0.022 |
| TG (mmol/L) | 1.070 | 1.277 | 0.803–2.030 | 0.301 |
| TyG index | 0.267 | 0.778 | 0.300–2.015 | 0.605 |
| CRP (mg/L) | 0.148 | 1.003 | 0.990–1.015 | 0.701 |
| Corrected serum calcium (mmol/L) | 2.798 | 2.655 | 0.846–8.336 | 0.094 |
| LVEF (%) | 0.559 | 0.980 | 0.928–1.034 | 0.455 |
| Calcification of abdominal aorta | 0.088 | 1.150 | 0.456–2.898 | 0.767 |
| Use of phosphorus binding agent | 2.170 | 0.530 | 0.227–1.234 | 0.141 |

**Notes.**

Variable assignment: Calcification of abdominal aortic: 1 = yes, 0 = no; Use of phosphorus binding agent: 1 = used, 0 = not used; Age, diastolic blood pressure, Hb, Alb, cystatin C, uric acid, TG, TyG index, CRP corrected serum calcium, and LVEF were raw data.

cardiovascular and cerebrovascular diseases in 36 cases (46.7%), infection in 29 cases (37.7%), gastrointestinal bleeding in nine cases (11.7%), and "other" in three cases (3.9%).

A comparison of survival rates between the CVC and non-CVC groups is shown in Fig. 3. During the follow-up period of six years, 30 patients (29.4%) died in the CVC group and 47 patients (19.5%) died in the non-CVC group ($P < 0.05$). A Kaplan Meier survival analysis showed that the overall survival rate of patients with cardiac valve calcification was lower than that of patients without cardiac valve calcification (Log rank test $\chi 2 = 183.803$, $P < 0.001$).

A comparison of CVC prevalence in patients with different primary diseases of CKD is shown in Fig. 4. The prevalence rates of CVC among patients with the most common three primary diseases in this study were each calculated. The prevalence rates of CVC in CKD patients with a primary disease of chronic glomerulonephritis, hypertensive nephropathy and diabetic nephropathy were 45.1%, 76.8% and 61.0%, respectively, with statistically significant differences between the three groups ($\chi^2 = 20.582$, $P < 0.001$).

## DISCUSSION

Cardiac valve calcification (CVC) is one of the most common complications in patients with CKD (*Bellasi et al., 2013*). Mitral and aortic valve calcification is very common in patients with CKD (*Plytzanopoulou et al., 2020*). In addition to valvular stenosis or regurgitation, CVC can also lead to cardiac conduction abnormalities and infective endocarditis (*Roberts, Salam & Roberts, 2022*). Mitral valve insufficiency and aortic stenosis are significantly associated with a decreased survival rate in CKD patients (*Marwick et al., 2019*). In most CKD patients, there is a long asymptomatic phase before the onset of clinical symptoms associated with valve calcification (*Willner et al., 2022*). Echocardiography is a sensitive
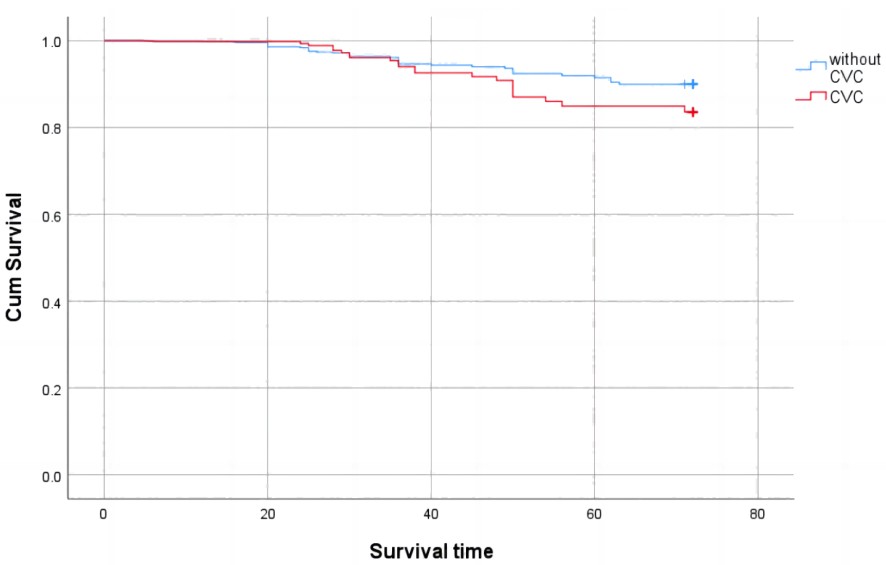

**Figure 3** Cumulative survival with all-cause mortality of CKD patients with valve calcification compared with those without valve calcification (Kaplan-Meier survival analysis).

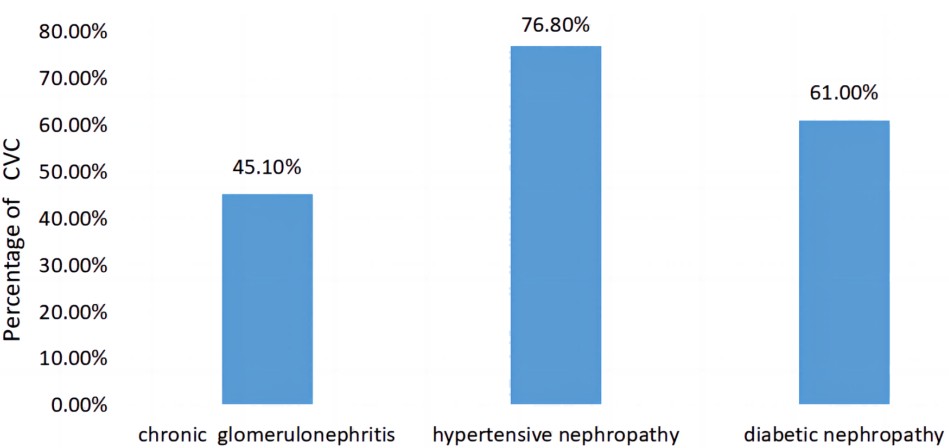

**Figure 4** Comparison of the prevalence of CVC in patients with different primary diseases of CKD.

and specific method to detect cardiac valve calcification, recommended by the KDIGO guidelines (*Kidney Disease: Improving Global Outcomes (KDIGO) CKD-MBD Work Group, 2009*).

The incidence of CVC in this study of 343 CKD patients was 29.7%, which is consistent with domestic research results (*Li et al., 2020*). Of the patients with CVC, 61.8% had aortic valve calcification, 20.6% had mitral valve calcification, and 17.6% had mitral valve calcification combined with aortic valve calcification. In accordance with the results of a previous study, aortic valve calcification had the highest incidence (*Cao et al., 2011*). Cardiac ultrasound is not routinely performed in healthy patients, so this study did not

include a healthy control group, but a previous study found that the risk of aortic valve calcification in CKD patients was 1.3 times higher than that in patients without CKD (*Fox et al., 2006*). The all-cause mortality and cardiovascular mortality of patients with aortic valve calcification are significantly higher than those of patients with mitral valve calcification, suggesting that aortic valve calcification contributes more to the risk of cardiovascular death (*Li et al., 2020*).

Cardiac valve calcification is the result of many factors, but the specific mechanism of CVC is not yet fully understood. In addition to traditional risk factors such as gender, age, hypertension, hyperglycemia and hyperlipidemia, new risk factors such as inflammation and malnutrition, calcium and phosphorus metabolism disorders, and hypomagnemia can promote its occurrence (*Stewart et al., 1997*; *Plytzanopoulou et al., 2020*; *Xiong et al., 2022*; *Ding et al., 2022*). Advanced age, higher serum albumin, higher cystatin C and lower uric acid levels were all associated with a higher risk of CVC. One previous study found that advanced age and low serum albumin/total albumin ratio were predictive indicators of cardiac valve calcification in hemodialysis patients (*Plytzanopoulou et al., 2020*). Advanced age is a recognized risk factor for CVC, both in the normal population and in patients with CKD (*Plytzanopoulou et al., 2022*; *Boon et al., 1997*). In this study, although the level of Alb in the CVC group was higher than that in the non-calcification group, but they both lower than the normal level of 40g/L, indicating that relatively elevated albumin levels was a risk factor for CVC.

Since the kidney is the only organ to clear plasma cystatin C, cystatin C levels can reflect kidney function more accurately and sensitively than other indicators (*Chew et al., 2008*). A regression analysis showed that a higher cystatin C level was a risk factor for valve calcification, suggesting that renal function damage is also a risk factor for CVC. It can also be seen in Fig. 2 that the incidence of CVC increases gradually with the progression of renal function damage, indicating that renal insufficiency may be a direct cause of CVC formation. The occurrence of CVC is caused by the cumulative effect of both known and unknown cardiovascular risk factors, which converge with renal impairment and reflect the underlying pathological process, such as atherosclerosis. CVC is also caused by some factors of renal insufficiency, such as increased inflammation, hypertension and mineral disorders (*Asselbergs et al., 2009*; *Raggi et al., 2002*; *Wang et al., 2001*). Uric acid is the end product of purine metabolism. In addition to promoting oxidation, uric acid also has antioxidant effects. The rise of uric acid in human evolution may be a protective factor against oxidative damage to the cardiovascular system because of its antioxidant effect (*Muraoka & Miura, 2003*). Oxidative stress is an important mechanism of endothelial dysfunction, which can promote the occurrence of cardiovascular diseases (*Sautin et al., 2007*).

Cardiovascular and cerebrovascular diseases are the leading causes of death in patients with end-stage renal disease (*Jankowski et al., 2021*; *Zong et al., 2016*). During the follow-up period of this study, the mortality rate of CKD patients was 22.4%, and the top three causes of death were cardiovascular and cerebrovascular diseases, infection, and gastrointestinal bleeding. A survival analysis showed that the survival rate of patients in the calcified heart

valve group was significantly lower than that in the non-calcified heart valve group, which is consistent with previous findings (*Bai et al., 2022*).

Although the mortality rate was higher in the CVC group than in the non-CVC group, there was no statistically significant difference in the causes of death between the two groups. The incidence of CVC in hypertensive nephropathy and diabetic nephropathy patients was higher than that in chronic glomerulonephritis patients (Fig. 4). Hypertension and diabetes are also associated with cardiovascular risk factors, which may aggravate cardiovascular calcification.

This study had limitations. The data used in this study are from a single center, and the sample size was limited. All the subjects included in the study were hospitalized CKD patients, most of whom had CKD stage 5, indicating selection bias, so this study did not include a representative sample of the general CKD population. In addition, since this is a retrospective study, a multicenter, large-sample prospective study is needed to further confirm the factors related to the clinical prognosis of CVC and non-CVC CKD patients identified in this study.

## CONCLUSION

In conclusion, the incidence of cardiac valve calcification is high in CKD patients, with aortic valve calcification as the most common type of CVC in this population. Advanced age, higher serum albumin and higher cystatin C levels were associated with a higher risk of CVC. Hyperuricemia was associated with a lower risk of CVC. The overall survival rate of patients with valve calcification was lower than that of patients without valve calcification. After six years of follow-up, 22.4% of CKD patients died. The top three causes of death were cardiovascular and cerebrovascular diseases, infection, and gastrointestinal bleeding.

### Funding
This work was supported by the General Project of the Second Affiliated Hospital of Anhui Medical University Clinical Research Cultivation Program of 2021 (2021LCYB20). The funders had no role in study design, data collection and analysis, decision to publish, or preparation of the manuscript.

### Grant Disclosures
The following grant information was disclosed by the authors:
General Project of the Second Affiliated Hospital of Anhui Medical University Clinical Research Cultivation Program of 2021: 2021LCYB20.

### Competing Interests
The authors declare there are no competing interests.

### Author Contributions
- Ju Wang performed the experiments, analyzed the data, prepared figures and/or tables, authored or reviewed drafts of the article, and approved the final draft.

- Jianping Xiao analyzed the data, prepared figures and/or tables, and approved the final draft.
- Ruifeng Wang performed the experiments, prepared figures and/or tables, and approved the final draft.
- Deguang Wang conceived and designed the experiments, prepared figures and/or tables, and approved the final draft.

## Clinical Trial Ethics

The following information was supplied relating to ethical approvals (*i.e.*, approving body and any reference numbers):

The Ethics Committee of the Second Affiliated Hospital of Anhui Medical University approved this study (YJ-YX2017-0040).

## Data Availability

The raw measurements are available as Supplemental File.

## Clinical Trial Registration

The following information was supplied regarding Clinical Trial registration:

No

## Supplemental Information

Supplemental information for this article can be found online at http://dx.doi.org/10.7717/peerj.15569#supplemental-information.

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
