# Peer review of "Influencing factors of cardiac valve calcification (CVC) in patients with chronic kidney disease and the impact of CVC on long-term prognosis: a single-center retrospective study"

_PeerJ, doi:10.7717/peerj.15569_

## Round 0.1 · original submission · Major Revisions

These reviewers have provided useful comments to improve the Article quality.

Reviewer 1 ·

Basic reporting

Thank you for the opportunity to review this manuscript.
In this study, the authors investigate the impact of cardiac valve calcification on the outcomes of patients with chronic kidney disease (CKD) among 343 CKD patients. The authors identified an association between the presence of heart valve calcifications and worse outcomes.

Experimental design

no comment

Validity of the findings

no comment

Additional comments

Major Comments
1. Were any non-CKD patients included in this study? Given this is a retrospective study of patients admitted to hospital, it is feasible that a control group could have been included in this study. This would be helpful in elucidating whether the changes identified in this study are related to CKD or are consistent across this population. This should be clarified and discussed in the manuscript.

Minor Comments
1. In the Introduction, the authors state “Cardiac valve calcification (CVC) can lead to cardiac conduction dysfunction, myocardial ischemia or 31 infarction, congestive heart failure and other complications, and increase the risk of cardiovascular death.” There is no mention here about valvular dysfunction which is a significant factor to miss in a list describing complications of heart valve calcification. This should be corrected.
2. One of the listed exclusion criteria is “heart valve”. I assume this is referring to patients with heart valve disease, but it does not make sense that patients with heart valve disease would be excluded from a study investigating heart valve calcification and CKD. The authors should clarify this exclusion criteria and why it was selected.
3. There are several acronyms used in the tables such as LVEF, CVC, iPTH, etc. These acronyms and others should be defined at the legend at the bottom of the tables.
4. Table 3 in the body of the manuscript has a title that is not in English. This should be corrected to be in line with the rest of the manuscript.
5. In line 164 the authors state “kidney can reflect kidney function more accurately”. It is unclear what is meant by this statement and it should be revised for clarity.
6. The tables and figures provided are appropriate and complement the information provided in the article.

Reviewer 2 ·

Basic reporting

The manuscript was written in clear, professional English. However, some minor improvements could be made:
Ln 39-40 – Please rephrase, this sentence is difficult to understand
Figure 3 – please remove Chinese signs under the CVC and give English one. What does it mean “censored” – please give explanation in figure footnote.
Table 3 – remove Chinese title, give English one, as in supplementary files
Ln 132 – delete doubled P values

The article was heavily under referenced. Multiple citations should be added to following sentences:
Ln 31 – “… cardiovascular deaths”
Ln 33 – “…. Increased mortality”
Ln 62 – please cite relevant work with validation of this formula
Ln 64 – as mentioned above
Ln 139-144 – each sentence need citation
Ln 165 – “…. sensitively” – citation is needed
Ln 187 – please cite relevant work
Ln 196 – please cite those studies.

Text of the manuscript has a professional, scientific structure. Professional background was provided. Results are relevant to hypothesis.

Experimental design

Ln 39-40 and ln 70-71 contradict each other. Ln 70-71 – I do not understand why did authors divide endpoint into more endpoints? Authors did not refer to these groups in the rest of the manuscript. Relevant analyses were not presented. From my opinion, the authors doubled their thoughts in Ln 71.

Ln 49 – “…Heart valve etc” and figure 1 – very vague exclusion criteria. It is unacceptable in scientific paper to report ambiguous exclusion criteria. Please list all these criteria, to make study replicable.

Research question was very broad, mostly answered by the results. Fills some gap in knowledge but is not a breakthrough. Good technical and statistical analysis, acceptable for publication.

Validity of the findings

Results and conclusions are in accordance with current literature. All underlying data have been provided. Conclusions answer the research questions, supported with generated results. Data have been provided (I recommend publication as supplementary file), I did not recalculate the results due to lack of IBM SPSS subscription.

Additional comments

Good, technical manuscript, suitable for publication.

Reviewer 3 ·

Basic reporting

• This paper is an interesting and insightful retrospective study of the influencing factors and impacts of cardiac valvular calcification among CKD patients. The manuscript requires a careful revision of written English; there are several instances where comprehension is affected, grammatical and punctuation errors require attention, and Table 3 title was still written in Mandarin. The English language should be improved to ensure that an international audience can clearly understand this manuscript; thus, it is suggested to contact a professional language editing service.
• This study’s consistency must be considered carefully. The title, abstract’s objective, and aims of this study described in the introduction must be consistent, as it will affect the study methods and results.
• The quality and resolution of the figures should be improved.

Experimental design

• This study is retrospective, but the initial included patients' characteristics should be properly noticed. Ideally, at initial patients’ inclusion in 2015, authors had to ensure that all recruited CKD patients did not have valve calcification. Those patients then should be followed up until this study ended, separated between CVC and no CVC groups, and then analyzed for CVC risk factors and outcomes. It is also recommended to perform sub-group analysis according to the CKD stages and etiologies.
• There is a loss to the follow-up group in this study. It is advised to state how this group's data was analyzed for study endpoints, and please also state this study's secondary endpoints.

Validity of the findings

• Looking at the raw data uploaded, there are many null data and probably several inappropriate data, such as extreme uric acid level, which is only 1-5 compared to hundreds. Authors should carefully complete and review the raw data before analyzing them.
• The baseline characteristics of CKD patients’ etiologies should be compared between CVC and no CVC groups to show whether there are possibilities of significant baseline differences.
• This study's top three mortality causes are cardio-cerebrovascular events, infection, and GI bleeding. These mortality causes should be compared between CVC and no CVC groups to show whether there are possibilities of significant baseline differences. It is also advised to discuss this finding in the discussion. Authors should explain further whether these mortality causes are associated with CVC or were only coincidences.

Additional comments

• In discussion, it is pertinent to provide further evidence, possible mechanisms, and reasoning of what makes hyperuricemia a protective factor, while high haemoglobin and albumin are risk factors, and also how CVC could be a protective factor for all-cause mortality among CKD patients.
• The conclusion might be considered to be separated from the discussion.
• The previous paragraph before Table 3 seems redundant with Table 3 itself.

Reviewer 4 ·

Basic reporting

The manuscript has two areas of weakness. First, lacks multiple references in all sections – ln33, ln 62, ln 187, ln 186 and lns 28-36. Secondly, manuscript requires proof reading in some areas – ln 40, Table 3 – remove Chinese signs, what does it mean “censored” on Figure 3? Please explain.

Experimental design

Please clarify what was the final endpoint – to many endpoints, only first part of sentence was utilized in the analysis (ln 40). Please provide precise exclusion criteria and give rationale for them. “Etc” in exclusion criteria is ominous sign for research transparency – authors could potentially pre-selected certain patients to get desired results.

Validity of the findings

Good technical manuscript. Results are in accordance with the literature. Thesis was answered by the results.

---

## Round 0.2 · accepted · Accept

This version is a good revised manuscript and can be accepted.

Reviewer 3 ·

Basic reporting

The language has been improved.

Experimental design

No additional comment.

Validity of the findings

No additional comment.

Additional comments

Thank you for your satisfying responses and significant improvements. The comments have been adequately addressed, and I would support the publication of this manuscript.